# Long-Term Follow-Up of 12 Patients Treated with Bilateral Pallidal Stimulation for Tardive Dystonia

**DOI:** 10.3390/life11060477

**Published:** 2021-05-24

**Authors:** Hiroshi Koyama, Hideo Mure, Ryoma Morigaki, Ryosuke Miyamoto, Kazuhisa Miyake, Taku Matsuda, Koji Fujita, Yuishin Izumi, Ryuji Kaji, Satoshi Goto, Yasushi Takagi

**Affiliations:** 1Department of Neurosurgery, Graduate School of Biomedical Sciences, Tokushima University, Tokushima 770-8503, Japan; koyama.hiroshi.jf@mail.hosp.go.jp (H.K.); morigaki.riyoma.1@tokushima-u.ac.jp (R.M.); med.miyake@tokushima-u.ac.jp (K.M.); matsuda.taku@tokushima-u.ac.jp (T.M.); ytakagi@tokushima-u.ac.jp (Y.T.); 2Center for Neuromodulation, Department of Neurosurgery, Kurashiki Heisei Hospital, Kurashiki 710-0826, Japan; 3Parkinson’s Disease and Dystonia Research Center, Tokushima University Hospital, Tokushima 770-8503, Japan; ryom@tokushima-u.ac.jp (R.M.); kfujita@tokushima-u.ac.jp (K.F.); yizumi@tokushima-u.ac.jp (Y.I.); sgoto@tokushima-u.ac.jp (S.G.); 4Department of Advanced Brain Research, Graduate School of Biomedical Sciences, Tokushima University, Tokushima 770-8503, Japan; 5Department of Neurology, Graduate School of Biomedical Sciences, Tokushima University, Tokushima 770-8503, Japan; rkaji@tokushima-u.ac.jp; 6National Hospital Organization Utano Hospital, Kyoto 616-8255, Japan; 7Department of Neurodegenerative Disorders Research, Graduate School of Biomedical Sciences, Tokushima University, Tokushima 770-8503, Japan

**Keywords:** tardive dystonia, deep brain stimulation, globus pallidus internus, long-term follow-up

## Abstract

Tardive dystonia (TD) is a side effect of prolonged dopamine receptor antagonist intake. TD can be a chronic disabling movement disorder despite medical treatment. We previously demonstrated successful outcomes in six patients with TD using deep brain stimulation (DBS); however, more patients are needed to better understand the efficacy of DBS for treating TD. We assessed the outcomes of 12 patients with TD who underwent globus pallidus internus (GPi) DBS by extending the follow-up period of previously reported patients and enrolling six additional patients. All patients were refractory to pharmacotherapy and were referred for surgical intervention by movement disorder neurologists. In all patients, DBS electrodes were implanted bilaterally within the GPi under general anesthesia. The mean ages at TD onset and surgery were 39.2 ± 12.3 years and 44.6 ± 12.3 years, respectively. The Burke–Fahn–Marsden Dystonia Rating Scale (BFMDRS) performed the preoperative and postoperative evaluations. The average BFMDRS improvement rate at 1 month postoperatively was 75.6 ± 27.6% (*p* < 0.001). Ten patients were assessed in the long term (78.0 ± 50.4 months after surgery), and the long-term BFMDRS improvement was 78.0 ± 20.4%. Two patients responded poorly to DBS. Both had a longer duration from TD onset to surgery and older age at surgery. A cognitive and psychiatric decline was observed in the oldest patients, while no such decline ware observed in the younger patients. In most patients with TD, GPi-DBS could be a beneficial therapeutic option for long-term relief of TD.

## 1. Introduction

Long-term treatment with dopamine receptor antagonists may have adverse consequences, such as the development of tardive syndrome (TDS), which includes tardive dyskinesia and tardive dystonia (TD) [1,2,3,4]. Both tardive movement disorders cause emotional and social distress; TD has a faster onset and is more painful, distressing, and disabling than tardive dyskinesia [2]. Clinical presentations of TD are usually similar to those of focal, segmental, or generalized primary dystonia. Cervical muscles are affected in two-thirds of all patients with TD [5,6]. Treatment of TD involves gradual withdrawal of TD-inducing medications, the substitution of atypical neuroleptics (e.g., clozapine) or the administration of tetrabenazine, dopamine agonists, and anticholinergics [4,7]. Pharmacological treatment of TD may be challenging and ineffective. Globus pallidus internus (GPi) deep brain stimulation (DBS) has been proven to be effective in medically refractory cases of primary generalized, segmental, or focal dystonia [8,9,10,11]. On reviewing the therapeutic perspective on TDS [3], we found that clinical similarity between TD and primary dystonia has led to the development of neurosurgical treatment modalities such GPi-DBS [8,12,13]. In 2008, we reported that GPi-DBS improved the dystonia motor score by 86% in six patients with TD at a median follow-up of 21 months [14]. To confirm the long-term efficacy and safety of GPi-DBS in patients with severe TD, we extended the follow-up of these six patients and studied six more patients. We assessed the clinical outcomes of GPi-DBS in a total of 12 patients with severe TD, 10 of whom were followed up for longer than 2 years (2–15 years) after surgery.

## 2. Materials and Methods

All patients included in this study fulfilled the diagnostic criteria for TD proposed by Adityanjee et al. [2]. To confirm that all inclusion criteria were fulfilled, the history and clinical characteristics of each patient were reviewed at the movement disorder team meetings consisting of at least one neurologist, one psychiatrist, and two neurosurgeons. We retrospectively retrieved the medical records of all patients with TD who received GPi-DBS at our hospital between January 2004 and December 2019. All patients included in this study provided written informed consent for the DBS surgery and follow-up examinations. This study was approved by the local ethical committee at Tokushima University. Bilateral GPi-DBS was performed as reported previously [14,15]. Under general anesthesia with propofol or desflurane, microelectrode recordings were performed for all patients. Quadripolar electrodes (model 3387, Medtronic, MN or 1.5 mm space, St. Jude Medical, MN) were implanted into the ventral margin of the GPi and connected to programmable pulse generators (Activa SC, Medtronic or Infinity, St. Jude Medical) implanted subcutaneously on the same day. Intraoperative microelectrode recordings and postoperative computed tomography confirmed appropriate electrode positioning. The electrode and electrical parameters providing the best improvement of dystonic symptoms with the least adverse effects were determined at three to four weeks after surgery. Neurological examination with the Burke–Fahn–Marsden Dystonia Rating Scale (BFMDRS) was performed preoperatively and postoperatively by the same rating investigator from the movement disorder team. To evaluate changes in BFMDRS from before surgery to 1 month and last follow-up period, statistical analyses were performed using the paired *t*-test. *p* values less than 0.05 were considered statistically significant.

## 3. Results

Twelve patients (seven men and five women) with severe TD were included in this study. Their characteristics are shown in Table 1 and Table 2. Although the data of six patients (patient number 1–6) have been previously reported, their data were updated by extending the follow-up period. Six additional patients (patient number 7–12) were enrolled. The mean age at the time of surgery of all patients was 44.6 ± 12.5 years. The mean neuroleptic exposure was 9.1 ± 8.6 years, and the mean age at TD onset was 39.2 ± 12.3 years. The mean duration between TD onset and surgery was 4.5 ± 4.2 years. Depression, schizophrenia, panic disorder, and bipolar disorder were underlying comorbidities, for which medications such as etizolam, sulpiride, and haloperidol were commonly prescribed. In 11 patients, TD affected the neck. The mouth and upper and lower extremities were affected in five patients. In all patients, optimal results were obtained with the following stimulator settings: mean amplitude of 2.8 ± 0.9 V/ 4.0 ± 0.4 mA (range, 1.7−4.4 V/ 3.5–4.2 mA), mean frequency of 94.5 ± 35.0 Hz (range, 60–130 Hz), and pulse width of 348.2 ± 163.0 μs (range, 60–450 μs) (see Table 2). 

The mean preoperative BFMDRS score was 35.3 ± 20.7 (n = 12, range, 12–75). All patients had improvement in BFMDRS scores postoperatively (Figure 1); the mean BFMDRS score was 9.7 ± 14.6 (n = 11, *p* < 0.001, paired *t*-test), and the mean improvement rate was 77.8% (range, 12%–100%) at 1 month after surgery. At the last follow-up (65.7 ± 54.1 months), the improvement was maintained with a 75% (n = 12, 12%−97%, *p* < 0.001) decrease for the total BFMDRS score (Appendix A). Individual changes in the BFMDRS score from baseline to 1 month and the last follow-up are shown in Appendix A. Ten patients were followed up for longer than 2 years (mean follow-up, 78 ± 50.4 months after surgery); the BFMDRS score was 8.9 ± 17.3 (n = 10, *p* < 0.0001, paired *t*-test), and the mean improvement rate was maintained in 80.4 ± 20.4% of them.

Preoperative and postoperative clinical features of a representative patient (patient number 9) are shown in the Supplementary videos. Adverse events of DBS surgery included postoperative wound infection and worsening mania, which improved with antibiotic therapy and adjustment of DBS parameters, respectively (Table 2). In two patients (patient number 10 and 11), GPi-DBS did not lead to sufficient beneficial effects. One patient (patient number 10) responded poorly to the stimulation and experienced worsened mania 1 month postoperatively. Another patient (patient number 11) had a BFMDRS score improvement of 65% at 1 month postoperatively; however, the efficacy of DBS gradually reduced, and the improvement was 25% at 46 months postoperatively. Both these patients had an older age at surgery (69 and 55 years old) and a longer duration between TD onset and surgery (19 and 9 years).

## 4. Discussion

We enrolled 12 patients with severe TD who underwent GPi-DBS at our hospital. Significant improvement in mean motor score was observed at both short- (75.6%) and long-term follow-up (>2 years, 78.0%) after GPi-DBS. Since 1980, 117 cases of TDS treated with DBS have been reported in 35 studies, many of which were single case reports. The 10 patients reported by Damier et al. in their case series exhibited a mean improvement of 55% ± 15.5% in the Abnormal Involuntary Movement Scale (AIMS) score at 6 months postoperatively [16]. A phase Ⅱ study reported improvement of more than 60% in both the AIMS and Extrapyramidal Symptoms Rating Scale scores for more than 6 years postoperatively and provided level C evidence of the American Academy of Neurology guidelines for the positive effects of GPi-DBS in patients with TDS. Our study included a larger number of patients than previous case series studies; this could contribute to the existing evidence on the beneficial effect of GPi-DBS on TD. In the present study, however, the BFMDRS score improvement rate in patient 10 was poor, and patient 11 had improved 1 month postoperatively but gradually worsened in the long term. Both had developed a mental illness after the age of 35, and patient 5 had been receiving antipsychotic drugs for over 30 years; these may be poor prognostic factors. Andrews et al. reported that, for all types of dystonia, shorter durations from onset to surgery were associated with better surgical outcomes [17]. However, according to the criteria proposed by the French Stimulation for Tardive Dyskinesia, DBS should only be considered in patients with persistent (>1 year) and severely disabling TDS treated with clozapine or tetrabenazine for at least 4 weeks. Our results emphasize the importance of considering surgical treatment for disabling TD refractory to medical treatment early in the disease course.

In this study, the optimal DBS parameters in four patients were long pulse width (180–450 μs) and low frequency (60 Hz). These stimulation parameters may be effective in patients with dystonia. However, the use of high- vs. low-frequency stimulation in dystonia has shown mixed results. While Alterman et al. suggested that the use of 60 Hz stimulation can be beneficial in some patients [18], another group preferred high-frequency stimulation [19]. Moro et al. concluded that high-amplitude and high-frequency stimulation might lead to better outcomes in patients with cervical dystonia [20]. Various pulse widths have been recommended for GPi-DBS. Coubes et al. recommended the use of 450 µs [21]. Another study did not show any significant difference between pulse widths of 60, 120, and 450 µs [22]. At present, because of these conflicting results, the stimulation parameters used during DBS depend to a large extent on personal experience. 

This report supports the effectiveness of GPi-DBS for treating severe TD refractory to best medical treatment. However, older age at the time of surgery and longer duration between TD onset and surgery might be poor prognostic factors for GPi-DBS.

## Figures and Tables

**Figure 1 life-11-00477-f001:**
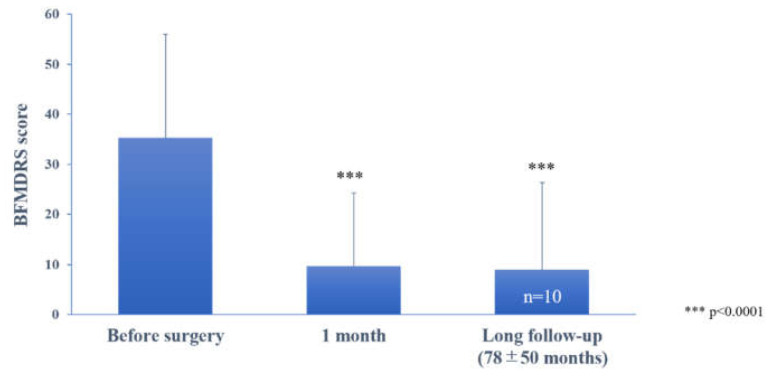
The effect of bilateral pallidal stimulation on tardive dystonia at 1 month postoperatively and at long-term postoperative follow-up (78 ± 50 months). Motor symptoms were assessed using the Burke–Fahn–Marsden Dystonia Rating Scale (BFMDRS). Vertical bars represent the standard error of the mean. *** *p* < 0.0001, paired *t*-test, compared with the preoperative score.

**Table 1 life-11-00477-t001:** Patient characteristics.

Patient Number	Age at Surgery	Gender	Age at TD Onset	Underlying Psychiatric Disease and Onset Age	Suspected Drugs	Drug Exposure (Years)	Duration Between TD Onset and DBS (Years)	Region on Dystonia
1	48	F	46	Depression, 40	Sulpiride	2	2	Eye, mouth, neck.
2	48	F	47	Bipolar disorder, 40	Tiapride	6	1	Neck, bilateral arms, left leg.
3	30	M	29	Schizophrenia, 28	Risperidone	1	0.5	Left arm, trunk, right leg.
4	47	F	40	Panic disorder, 31	Perphenazine	5	7	Neck, trunk.
5	39	M	39	Depression, 36	Perphenazine	0.5	0.5	Neck, right arm, right leg.
6	55	M	NA	Anxiety neurosis, 51	Haloperidol	4		NA
7	42	M	41	Depression, 33	Sulpiride	7	1	Neck, mouth, bilateral arms.
8	30	F	19	Schizophrenia, 15	Risperidone Haloperidol	14	11	Eye, mouth, neck, bilateral legs.
9	25	M	19	Schizophrenia, 14	Risperidone Haloperidol	11	6	Neck, trunk, bilateral legs.
10	69	M	59	Bipolar disorder, 37	Aripiprazole Quetiapine	32	10	Neck, trunk
11	55	F	44	Schizophrenia, 43	Risperidone Haloperidol	10	9	Eye, mouth, neck, trunk
12	49	M	48	Schizophrenia, 31	Risperidone	17	1	Neck, mouth, bilateral arms.

NA = not available.

**Table 2 life-11-00477-t002:** The outcomes at baseline, 1 month and long-follow up after GPi-DBS.

Patient Number	BFMDRS Baseline	BFMDRS 1 Month after DBS	Improvement Rate at 1 Month (%)	BFMDRS at Last Follow-Up	Improvement Rate at Last Follow-Up (%)	Postoperative Follow-Up Time (Months)	Stimulation Parameters with Best Response Amplitude/Frequency (Hz)/Pulse width (μs)
1	26	NA	NA	3	88.5	50	Bilateral 3(+)2(−) 4.4V/60/450
2	21	8	61.9	2	90.5	88	Rt. Case(+) 1(−) 2.0V/130/450 Lt. Case(+)2(−) 2.5V/130/450
3	19	4	78.9	2	89.5	107	Rt. Case(+)2(−) 2.8V/130/450 Lt. Case(+)2(−) 2.5V/130/450
4	32	0	100.0	4	87.5	186	Rt. Case(+)0(−)1(−)2(−) 1.6V/130/450 Lt. Case(+)1(−)2(−) 1.8V/130/450
5	12	1	91.7	3.5	70.8	127	Bilateral Case(+)1(−) 2.0V/130/60
6	75	9	88.0	9	88.0	3	NA
7	35	7.5	78.6	7	80.0	72	Rt. Case(+)1(−) 2.5V/60/450 Lt. Case(+)1(−) 2.5V/60/360
8	26	1	96.2	3	88.5	24	Bilateral Case(+)2(−) 4.2 mA/130/60
9	29.5	1	96.6	1	96.6	56	Rt. Case(+)1(−) 3.5V/60/450 Lt. Case(+)1(−) 3.8V/60/450
10	51	45	11.8	45	11.8	6	Bilateral Case(+)1(−) 3.5 mA/60/450
11	74	26	64.9	55	25.7	46	Bilateral Case(+)1(−)2(−) 3.0/60/180
12	23	3	87.0	3	87.0	24	Rt. Case(+)1(−)2(−) 4.1 mA/90/400 Lt. Case(+)2(−)4.4 mA/90/450

BFMDRS = Burke–Fahn–Marsden Dystonia rating scale; NA = not available.

## Data Availability

The data presented in this study are available on request from the corresponding author. The data are not publicly available due to patient privacy considerations.

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
