# Peer review of "Long-Term Follow-Up of 12 Patients Treated with Bilateral Pallidal Stimulation for Tardive Dystonia"

_life, 2021, doi:10.3390/life11060477_

Round 1

Reviewer 1 Report

In this study, Koyama and colleagues investigated the long-term efficacy of bilateral GPi DBS in treating tardive dystonia by comparing Burk-Fahn-Marsden Dystonia Rating Scale (BRMDRS) outcome measures one-month post op, to 2-15 years post op. They report TD improvement in all patients following GPi DBS at one month and long-term follow up. A few patients got worse between one month and long-term follow-ups but remained improved compared to pre-surgery measures. Perhaps most notable, the oldest patients (at the time of surgery) had the worse outcomes. This study employed a simple design, only reporting BRMDRS as an outcome measure. This simple design limits the conclusions that can be drawn, but the authors did not speculate beyond the results and presented the findings clearly. I believe this study is of interest to clinicians and scientists working with TD patients and DBS in general. I have some minor comments that I think should be addressed before publication.

  • Line 21 – I suggest changing “these many patients” to “more patients are needed to better understand the efficacy…”
  • Line 32 – I suggest rewording this sentence. Perhaps (if this was patients 10) “Cognitive and psychiatric decline was observed in the oldest patient, while no such declines were observed in the younger patients.”
  • Methods – pleases describe the statistical tests performed to establish significant BRMDRS reductions from before surgery to 1 month and long flow-up periods.
  • Formatting – I’m not sure, but I believe periods should go to the right of the reference. E.g., “[1].” Not” .[1]”
  • Comment – it would have been interesting if you included a questionnaire or interview patients about their quality of life / subjective improvement. This would provide valuable information that would complement the BRMDRS scores. Something to consider for future experiments.
  • Line 123 – please consider adding “(10,11)” such as “In two patients (10,11),…”
  • Line 131 – I suggest changing the word “encountered” to “enrolled”, or “studied” or “investigated”
  • Line 139 – a description of what level C evidence is would be helpful
  • Discussion – a clearer description of the number of patients who got worse from one month to long term is needed.

Reviewer 2 Report

this paper is very interesting and results may be useful in clnical practice 

Author Response

We thank you and the reviewers for your thoughtful suggestions and insights. The manuscript has benefited from these insightful suggestions. I look forward to working with you and the reviewers to move this manuscript closer to publication in Life.

Reviewer 3 Report

Major Issues

Abstract and results section results do not match. In the abstract, it is stated that at one-month post-DBS there was a 75.6% improvement, and at 78months there was a 78% improvement. However, in the results section is stated that there was a 76.8% improvement at one month, and at 66 months there was a 75.4% improvement. It would be helpful to clearly list the number of patients included in each analysis with the level of significance for the paired t-test.

The Burke-Fahn-Marsden Dystonia Rating Scale is the only clinical outcome used to describe clinical effects of DBS in this work. For the BFMDRS, the score is based on factors that include provoking and severity issues. It would be nice to know what aspects of these factors were reduced by DBS and which appear to be resistant. For example, I think readers would be interested to know what the impact of DBS was on the “general” provoking factor as well as the “severity” factor of neck. Additionally, do the changes in certain factors change at the earlier or later visit? A summary, table or figure to describe what specific changes were observed in this cohort is needed to better explain the clinical improvements.

A description of the BFMDRS data collection is needed with appropriate reference to the rating scale. Was this person experienced in rating dystonia? Was the same person assessing patients before and after surgery?

Minor Issues

The results section mentions that two patients had poor outcomes (10 and 11, line 124-127). However, the discussion section mentioned that two different patients had poor outcomes (4 and 5, line 143). Please clarify in the text if the authors feel that 4 people had poor outcomes and edit to have the results and discussion match for consistency. As is, I believe that perhaps 4 people had poor outcomes but am confused.

Figure 1: Unable to read y-axis labels clearly. Due to the small sample size, data may be best plotted as single datapoints in different colors to see the individual changes, overlaid with the mean and SEM. As plotted, information regarding the variance in changes within and between subjects is lost. Since this paper is a case-series, I think it would be of interest to readers to see this data visually.  

In video S2: the study participants eye makes it out of the blurred section on the upper left corner. It would be best practice to blur this out if possible.
